# Role of Palliative Care in Onco-Hematology Retrospective Observational Cohort Study in Deceased In-Hospital Patients with SACT at the End of Life: Experience with Real-World Data from a Cancer Monographic Institution

**DOI:** 10.3390/cancers17213467

**Published:** 2025-10-28

**Authors:** Lourdes Pétriz, Esther Asensio, Eva Loureiro, Joan Muniesa, Gala Serrano, Tarsila Ferro

**Affiliations:** 1Strategy and Quality Department, Institut Català d’Oncologia, L’Hospitalet de Llobregat, 08908 Barcelona, Spain; easensio@iconcologia.net (E.A.); tferro@iconcologia.net (T.F.); 2Computer Systems Service, Institut Català d’Oncologia, L’Hospitalet de Llobregat, 08908 Barcelona, Spain; eloureiro@iconcologia.net (E.L.); jmuniesa@iconcologia.net (J.M.); 3Palliative Care Department, Institut Català d’Oncologia, L’Hospitalet de Llobregat, 08908 Barcelona, Spain; gala@iconcologia.net

**Keywords:** cancer, solid tumors, hematological malignancies, end of life, palliative care, chemotherapy, mortality

## Abstract

**Simple Summary:**

This study aims to provide a better understanding of the profile of patients referred to palliative care (PC) and, also, what PC provides them. We used the Mortality Subcommittee database, which records hospital deaths of patients who have received SACT in their last month of life. We complemented the information of the registry with some variables of palliative care activity. We present here the comparative results between patients included or not included in Palliative Care Program (PCP) and between pathologies: solid tumor (ST) vs. hematological malignancies (HM). We found important differences between ST and HM, showing that they are very different pathologies and require different palliative approaches. Finally, we put forward some recommendations.

**Abstract:**

Background: The American Society of Clinical Oncology (ASCO) established recommendations for palliative care (PC), and they still remain the most trusted source overall. The standard published by C. Earle (defined in solid tumors) for referral to PC is > 55%. However, these rates remain unclear in general onco-hematology. Our referral rate reaches 60%; while it meets the standard, there are significant differences between ST and HM. Several authors have already pointed out these discrepancies. Arguing in some cases its possible relationship with the different behavior of professionals with different pathologies. Objective: The primary objective of this work is to understand the role that PC plays in onco-hematology and to determine the profile of patients referred to PC. Therefore, the article aims to establish some recommendations related to the results of prevalent characteristics. Methods: The Mortality Subcommittee (MS) includes and registers in a database all cancer patients who died in hospital undergoing systemic anticancer therapy (SACT) in their last 30 days of life (SACT ≤ 30 d). PC, in turn, works on relieving symptoms related to the disease and the patient. To understand the impact of PC in the MS database patients, we reviewed the literature for symptoms related to palliative care activity. Subsequently, we selected some signs and symptoms, by consensus with our PC specialists, in order to add them to the MS database and register them retrospectively. We measured the percentage of patients who registered these symptoms based on the data found in their electronic records. The results include the comparison by group: between patients referred or not to the PC program (PCP), and between the pathologies ST and HM. We used the programming language R (version 4.2) in our statistical analysis, including the “compareGroups” package (version 4.6), applying the pertinent tests based on the distribution of the data. Results: We completed the records on the 1681 patients from the period 2020–2023. 59.4% were men, the average age was 65.5 years, and 73.5% had ST and 26.5% had HM. Patients with lung cancer predominate (28.5%), with 71% of them being in the stage IV, followed by leukemia (9.76%). 60% are in progression of their disease, and 77% have advanced disease (AD). The average therapeutic aggressiveness indicators were SACT < 30 d: 38.9% (ST: 33.4%; HM: 70.97%); SACT < 14 d: 16.36% (ST: 13.76%; HM: 31.56%); the change in therapeutic regimen was 22% (ST: 20.8%; HM: 25.1%). The referral rate to PCP was 59.7% (ST: 68.2% and HM: 36.3%). Late referral (PCP ≤3 days before death) occurred in 29.2% of all patients, being 29% for ST cases and 30.4% for HM cases. Regarding the recording of signs and symptoms: psycho-emotional and analgesia regimens (including opioids) are better recorded in the PCP group (*p* < 0.001); the more physical symptoms (dyspnea, bleeding, infections, and severe symptoms) do not present statistically significant differences, although the severe symptoms in the PCP group are more disabling (cerebral involvement, spinal cord compression, vertebral crushing). The number of bags of blood products transfused is significantly lower in the PCP group (average 6.9 vs. 12.7). The total number of symptom variables with significant statistical differences was 13 for ST and 8 for HM. Conclusions: In this cohort, patients visited by PC had a better record of psycho-emotional symptoms. We consider that patients who are in any of the following situations should be referred to PC: initial diagnosis of stage IV lung cancer, leukemia; patients with advanced disease; presence of pain requiring opioids; psychoemotional symptoms; need for >7 to 15 transfusions of blood products and, if there are disabling symptoms. PC improves professional interest in the psycho-emotional and fragility situation of these patients. According to our data (in terms of the number of variables with significant differences by pathology group), we observed that hematologists tend to take on palliative tasks more frequently than their oncologist peers, who delegate them to PC in order to have more time dedicated to their specific field.

## 1. Introduction

### 1.1. Background

Since 2012, the American Society of Clinical Oncology (ASCO) has recommended the use of palliative care (PC) in patients with TS from the moment of diagnosis of metastatic cancer and/or with a high symptom burden. In the 2016 update, PC is defined as “care focused on the patient and their family, optimizing quality of life with anticipation, prevention and treatment of suffering” [1,2]. Similar is the WHO definition: “palliative care as the prevention and relief of suffering of adult and pediatric patients and their families facing the problems associated with life-threatening illness. These problems include physical, psychological, social and spiritual suffering of patients and psychological, social and spiritual suffering of family members” [3]. The International Association for Hospice and Palliative Care developed a consensus-based definition of palliative care (PC) that focuses on the relief of serious health-related suffering, a concept put forward by the Lancet Commission Global Access to Palliative Care and Pain Relief in 2020 [4,5]. In 2024, it extended the care recommendation to address the physical, intellectual, emotional, social and spiritual needs of patients. It proposes the use of PC throughout the oncological disease process. The inclusion of patients in clinical trials in PC is also beneficial [1,6,7].

According to published indicators of therapeutic aggressiveness and referral of patients to hospice [8,9,10,11,12,13], a standard of good practice is described when referral to hospice is >55%. We assume the terms hospice referral and Palliative Care Program (PCP) referral are equivalent [14].

This study is based on the work of the Mortality Subcommittee (MS) of the “*Institut Català d’Oncologia*” (ICO). Since 2017, the MS has systematically analyzed and recorded all in-hospital deaths of adult patients with cancer between SACT < 30 days, and monitored these indicators annually. The data come from three centers within the same institution, which treat between 45% and 50% of cancer cases in Catalonia. Regarding the indicators of referral to PC between ST and HM, we observed that their differences over time remained within a range similar to that published, despite our cohort only including hospital deaths [15,16]. The goal was to know if the differences are due to the attitude of the professionals (oncologists and hematologists) like Wedding, Auret, Odejide, and Leblanc described [17,18,19,20], or if the PC service plays a different role [21]. After discussing the topic with our PC specialists, we proposed a retrospective review of some signs and symptoms related to their practice, using the available cases in the MS database to shed light on the role they play in our onco-hematology patients.

### 1.2. Objectives

The main objective is to understand the role that PC plays in supporting each group of pathologies. The indirect indicator of PC activity is the measure of the percentage of symptoms identified in electronic records (study with real-world data).

According to the prevalence obtained and bibliographic support, we intend to make some recommendations for the referral of these patients to PC.

## 2. Materials and Methods

### 2.1. Study Design

From the systematic work of the MS, we observed that, from its beginning in 2017 until 2023, of the 7586 patients who died in the hospital, 2737 underwent SACT in their last 30 days of life and registered in the database.

Utilizing the resources of the database and jointly reviewing the signs and symptoms associated with PC activity, we decided to conduct a small-scale project, selecting and adding 25 new variables to the database registration form.

The variables were selected based on those in our PC form and some published variables [20,22,23,24]. This involves a series of questions about the presence or absence of a specific element and, when documented, whether it is positive or negative.

We proposed a retrospective review of these signs and symptoms, starting with the most recent cases. We obtained the information from electronic medical records, completing the forms manually.

Due to resource limitations, we were only able to complete the 1681 records corresponding to the period: January 2020 to December 2023 (61.2% of the total records in the complete database). The descriptive analysis and results focus on this descriptive and retrospective cohort of 1681 consecutive cases (Figure 1).

We performed a group comparison between those included and not included in the Palliative Care Program (PCP), and for ST vs. HM.

Scope: The data correspond to three cancer centers of the same institution, with different geographical locations. Data collection was manual by a qualified data recorder and supervised by an oncologist.

We have not included a multivariate analysis because we designed the study as a descriptive observational analysis, focused on descriptive comparisons for methodological reasons: description of the differences between referrals to PCP and between pathologies (TS vs. HM) in the real world and the retrospective nature.

On the other hand, to determine if the conclusions would be applicable to the entire database, we compared characteristics between the two periods: with and without symptom registration, showing no significant differences (see Appendix A).

### 2.2. Participants

Inclusion criteria: adults ≥18 years; diagnosis of any type of cancer (ST and HM); in-hospital deaths of patients receiving SACT ≤ 30 days; and have information in the module about new variables related to palliative care activity. Inclusion period: January 2020 to December 2023. Total number of patients included: 1681 patients.

### 2.3. Variables

For the selection of variables, we reviewed, on the one hand, the form used by our PC services and, on the other, in the literature [25,26,27]. The set of signs and symptoms that seemed appropriate and adjusted to our resources was 25 variables. These refer to symptoms (physical and/or psycho-emotional), use of pain relievers (including opioids) and blood products, signs of infection and bleeding, laboratory data (albumin, LDH, hemoglobin and platelets), and refer to the psychosocial program (PSOP).

The variables collect information about the following list of questions:

Is there an evaluation of pain? Is there an analgesic regimen prior to admission? Is there an analgesic regimen upon admission? Does the analgesic regimen include opioids before admission? Does the analgesic regimen include opioids upon admission? Is there an evaluation of dyspnea? Is there an evaluation of insomnia? Is there an evaluation for asthenia? Is there depression screening? Is there an evaluation of depression with the “ENDICOTT” test? Is there an evaluation of sadness? Is there an evaluation of anxiety? Does the patient have severe symptoms? What are the severe symptoms? Other severe symptoms: specify which ones. Was intubation performed? Did the patient have an advanced disease? Date of advanced disease. Did they receive transfusions? How many? Did they suffer from infections? Did they suffer from bleeding? Did they refer to the psychosocial program?

Laboratory: Platelet count (×10^9^/L), LDH level (uKat/L)

Laboratory data correspond to the date before the administration of the last SACT. The data on hemoglobin and albumin are already available in the SM database.

We included the question about sadness (which we do not find in the literature) because it appears on our institution’s PCP form, since our CP professionals consider sadness to be an independent symptom, not always related to anxiety or depression.

Refer to the psychosocial program already appearing in the SM database.

The number of transfusions of blood products is the total since the diagnosis of the disease. For severe symptoms, a dictionary was established, and those not included in the dictionary but considered equally severe were recorded as “other”. 

### 2.4. Data Sources/Measurement 

In order to define advanced disease (AD) and its time of onset, we developed a decision-making algorithm with the variables and their dates of initial stage (I, II, III and IV, for most tumors), progression (the first recorded) and the “prognostic score” (for leukemia) as relevant for each tumor.

To study time curves, we determine the intervals between dates of relevant events. We used “time of AD definition”, “inclusion in the PCP” (registered for the first time to appear in the electronic medical record), and death.

### 2.5. Bias

All consecutive cases from the 2020–2023 period met the criteria and were included, thus minimizing selection bias.

The retrospective nature of data collection from electronic records can introduce biases in the information; therefore, the trained personnel who performed the data entry manually work under medical supervision.

The fact that all cases correspond exclusively to hospital deaths could also represent a bias. In this sense, we accept that our indicators may have values higher than those published by other authors. Hospitalization indicates a condition of greater severity or risk for patients.

The definition of AD in hematology is more complex than in ST. This is especially true in the case of leukemia [28]. This complexity also partially limited our ability to obtain AD-PCP intervals for all HM.

The inclusion of all ages, all causes of death, and patients with advanced and non-advanced disease may represent a bias in comparison with other published data.

Clinical data comes from the electronic medical records of real patients. This is not data from epidemiological records or administrative billing, nor synthetic data.

Resource limitations have influenced the lack of continuous monitoring of signs and symptoms and, therefore, the evaluation of the appropriate moment for referral to palliative care. It has also limited the ability to establish the correlation with psychosocial and quality-of-life variables, which is why we have left them out of our scope of study.

The new variable registration only considers whether the variable is registered in the electronic medical record of each patient. Therefore, we assume that we only have an indirect measure of PCP activity.

Based on the results obtained, our recommendations have to do with the most prevalent groups in the cohort and the differences in the recording of symptoms (and other clinical factors) between patients referred to palliative care and those not referred.

### 2.6. Study Size 

The sample size corresponds to the complete period: 2020–2023, due to the availability of the new variables. We did not perform formal power calculations because this is an exploratory analysis of systematically collected real-world data.

### 2.7. Statistical Methods 

We used the R programming language and a tool developed within Shiny to facilitate descriptive and time curve analyses.

The R package “compareGroups” (Package source: version 4.6) [29,30] is the tool used for comparison between groups. This package allows the creation of tables and the display of the results of univariate analysis (stratified or not) and groups of categorical variables. For continuous variables, we have used the “Shapiro–Wilk test” to check normality and decide whether the variable follows a normal distribution or not. In the case of normality, the mean and standard deviation are calculated, and we perform the “*t*” or “ANOVA” test to compare between groups. In case of non-normality, the median and the “Kruskall-Wallis” test are calculated. For categorical variables, relative and absolute frequencies are calculated, and the “Fisher’s exact” test is calculated when the expected frequencies are less than 5%.

For time-to-event analysis, we used survival curves with the “compareGroups” package, which implements Kaplan–Meier estimators for survival data. The package calculates median survival times with confidence intervals and performs log-rank tests to compare survival distributions between groups. We determined intervals between relevant dates: advanced disease (AD) definition, PCP inclusion (first recorded appearance in electronic medical record), and death. Survival comparisons were performed between pathology groups (ST vs. HM) and PCP inclusion status, using time from AD definition to death as the primary endpoint. 

The temporal stability analysis compared baseline characteristics between periods: without enhanced data collection (first: 2017–2019) and with data collected (second: 2020–2023), using available variables (Appendix A). The results of the study focus on the cohort from the second period.

## 3. Results

### 3.1. Limitations

Limited resources led to the exclusion of the 2017–2019 period from this study, as it did not contain data on the new variables.

### 3.2. Results in This Retrospective Study Cohort

#### 3.2.1. General Descriptive

In this study cohort, 59.4% are male; average age is 65.5 years (55.9% being over 65 years old); 73.5% are ST and 26.5% are HM. The most common tumor pathologies are lung cancer (28.5%), leukemia (9.76%), non-Hodgkin lymphoma (8.63%), and breast cancer (7.02%). The sum of cases of digestive cancer (upper and lower) is 9.64% but in Figure 2, they appear separately.

Figure 3 shows the distribution of tumors by stages, with a predominance of patients with stage IV lung cancer. Of the 315 cases of stage IV lung cancer, 218 (69.2%) were referred to PCP and 97 (30.8%) were not. In contrast, for the 158 patients with leukemia, 42 (26.6%) were referred to PCP and 116 (73.4%) were not. For more details on the distribution of patients referred to PCP according to tumor type, see Appendix A.

#### 3.2.2. Outcome Data

Almost 77.8% of patients have advanced disease (AD) (ST: 87.9% and HM: 50%, *p* < 0.001). 84.8% present comorbidities (mean: 3.3 and median 3) of those who present comorbidities, 60% of patients present more than two comorbidities (ST: 58.676% and HN: 64.24%). They made a change in therapeutic regimen of 22% (ST: 20.8% and HM: 25.1%). Hospital stays longer than 14 days (in the last admission) occurred in 28.2% (ST: 22.8% and HM: 43%, *p* < 0.001).

The SACT between ≤14 days rate is 16.36% (ST: 13.76% and HM: 31.56%, *p* < 0.001). 

The PCP referral rate was 59.7% (ST: 68.2% and HM: 36.3%, *p* < 0.001). PCP ≤ 3 d rate: 29.2% (ST: 29% and HM: 30.4%).

The tumor types included in PCP, by percent, were as follows: lung (32.3%); all digestives (10.07%); breast (8.76%); gynecological and urological (5.68%); pancreas (5.18%); head and neck (5.08%); NHL (5.38%); leukemia (4.18%); and myelomas (4.38%) (Appendix A). 

In this cohort, it also stands out the following (Appendix A):-Death at the debut of the disease (first cycle of the first treatment) occurred in 20.9% (351 patients): 124 of them (35.4%) had lung cancer (80.6% in stage IV), and 56 had leukemia (71.4% acute leukemia).-The place of treatment administration was a hospitalization room in 26.7% of patients (HM increases to 51.6% and ST decreases to 17.8%); day hospital by 59.4% (for TS, it was 68.7% vs. 33.4% for HM) and outpatient was 13.7%, similar in both groups (15% for HM and 13.2% for ST) (*p* < 0.001).-The last SACT administered was part of a clinical trial in 8.06% (ST: 6.4% and HM: 12.7%, *p* < 0.001)-The hospital place of death was, for the most part, in acute hospitalization units:
Onco-hematology rooms: ST: 65.1%/HM: 56.2%;Other acute hospital units: ST: 5.3%/HM: 7.6%.ICU: ST: 5.84%/HM: 25.3%;Emergencies: ST: 1.7%/HM: 0.9%.Palliative Care Room: ST: 21.7%/HM: 9.93%.

#### 3.2.3. Main Results

Table 1 shows the details of the data recording of clinical factors, signs, and symptoms of the entire cohort and compared the “No PCP” and “Yes PCP” groups. In the last column, we observe the level of statistical significance of each variable. The added detail “log” at the end of the name in all symptoms means that there is logged data related to that symptom. Adding “Yes” after the variable means that the record has a positive value.

In the PCP group, the clinical factors show that the patients are younger, the predominant pre SACT ECOG is 2–4, they have a lower BMI, a higher proportion of AD, less debut in therapeutic scheme, with greater recording of level therapeutic intervention (LTI), greater coherence between the recorded LTI and medical action, fewer ICU admissions, and fewer intubations. For signs and symptoms, the psycho-emotional present significant differences, while for physical symptoms, there are no differences in the recording. For laboratory data, hemoglobin and platelets only approach significance.

Detailed symptom analysis by PCP referral: We correlated the variables with each type of tumor and observed some variations that we present below (see Appendix A).
(a)Pain management: registration globally occurs in 98.71%, being positive in 82% of cases. Of the positive cases, 56.9% received opioids before their last hospitalization, and this rose to 79.8% during their last hospitalization. By tumor type, the prevalence of pain registration (in absolute numbers) has been lung, leukemia, NHL, digestive, gynecological, head and neck, and myeloma. However, in percentages (with respect to the total number of cases of each same tumor), the positive records for pain were similar for all types of tumors and their value ranges between 80% and 90% (see Appendix A).(b)Respiratory symptoms: For dyspnea, the differences between groups are also significant. Dyspnea registration: PCP 95.94% vs. non-PCP 90.94% (*p* < 0.001). Positive dyspnea records: PCP 62.5% vs. non-PCP 61.0%.

By tumor type, the registration prevalence, in absolute values, is lung (36%), breast, leukemia, NHL, digestive, head and neck, and gynecological. While in percentage of the same type of tumor, the registration in lung cancer amounts to 78.5%, followed by breast, CUO (carcinoma of unknown origin) and digestive, around 70% in these cases.
(c)Psycho-emotional symptoms: Psycho-emotional symptoms (insomnia, anxiety, sadness, and depression) double and even triple their records and the positive register, when patients are in the PCP group. For anxiety registration: PCP 68.7% vs. non-PCP 34.6% (*p* < 0.001). Positive anxiety records: PCP 54.6% vs. non-PCP 29.8%. Appendix A shows the differences by type of tumor.(d)Severe symptoms: The percentage of severe symptoms recorded did not show significant differences. In the Yes-PCP group, it was 36.9% vs. 33.5% in the No-PCP group. However, we observed differences in distribution: the Yes-PCP group had a higher proportion of disabling complications (brain metastases, spinal cord compression, and vertebral crush). Table 1 shows the complete list of symptoms recorded in each group.

The PCP group had fewer transfusions: mean 6.91 vs. 12.7 bags, and had lower infection rates.

In Table 1, we also see significant differences (*p* < 0.001) by PCP groups yes/no in the following other parameters: age, ECOG pre SACT, BMI (but not by categories), disease progression, AE, therapeutic regimen (only in debut), ICU admissions and intubations.

Our center uses a level of therapeutic intervention (LTI) scale [31] to regulate the patient’s therapeutic needs. The recommendation for LTI registration is upon admission, after an evaluation and within the first 24–48 h. This is so that any professional can act according to the evaluation. This element is part of the variables in the SM database, as well as the coherence of the professionals’ actions with respect to what LTI recorded. We have included the values obtained in Table 1 since they were also statistically significant in the comparison of PCP groups.

Regarding referrals to PSOP, we can see that 100% of PCP patients are also in PSOP. On the contrary, 89% of the patients in PSOP are also in PCP (*p* < 0.001). Pathology groups referred to PSOP 73.8% of ST vs. 49.5% of HM. 

In Table 2, when comparing the new variables between ST and HM, and the subgroups of being or not on PCP, we observed there are more variables with statistically significant differences in ST than in HM (13 vs. 8). 

Pre-advanced disease consultations: When calculating the interval between referral to the PCP and determination of AD status, we observed negative intervals. These negative intervals indicate PC consultations made before AD determination. Figure 4 shows variations according to tumor type. The types with pre-advanced visits correspond with higher prevalence to lung, myeloma, head and neck, breast and gynecological cancer. In view of these data, we looked for the possible correlation between the most prevalent symptoms recorded in each type of tumor and the presence or absence of a palliative specialist in the tumor committees.

In cases with a palliative specialist on the committee (lung, myeloma, head and neck), the most frequently recorded symptom was pain; for breast and gynecological cancer with the absence of a palliative specialist in the tumor committees, there was a greater record of psycho-emotional symptoms. 

## 4. Discussion

### 4.1. Key Results 

In general, there are better rates of recording symptoms, especially psycho-emotional ones, when patients were referred to the PCP. The registration of vital symptoms was similar (Table 1). It is evident that patients referred to PCP receive better care in psycho-emotional aspects. Globally, we can say that care in PC reaches a good level of quality of care in patients with TS but not in MH.

### 4.2. Definition of AD in Hematology

The definition of AD in hematology is more complex than in TS. This complexity limited our ability to calculate accurate AD-PCP intervals for all hematologic malignancy registries. 

### 4.3. Management Differences by Specialty

The disparity between solid tumor and hematological neoplasm PCP referral rates (68.2% vs. 36.3%) may reflect persistent reluctance among hematologists to refer patients, as well as observing that the management of these patients by palliative physicians is similar between both groups of pathologies.

This is consistent with the literature describing hematologists’ preference for maintaining relationships with their patients and concerns about communicating prognosis. The habit of hematologists following their patients until the end of life is still present. They aim to prevent the patient from losing their life expectancy or feeling abandoned by the hematologist, with an error in the concept of identifying palliation only with “end of life” instead of “support” and for fear of having different opinions if there are more doctors to consult [25].

As Hui [32] reported in a survey of 240 oncologists and hematologists (with a response rate >70%), hematologists are more likely to prescribe chemotherapy in patients with PS 4 and 1-month overall survival (*p* < 0.001) and are less comfortable than oncologists talking about death (72% vs. 88%, *p* = 0.007) and referring patients to palliative care. (81% vs. 93%, *p* = 0.02).

In Table 2, when comparing the new variables between ST and HM, and the subgroups of being or not being on PCP, we see that there are more variables with statistically significant differences in ST than in HM (13 vs. 8). The variables that are not in HM are analgesia before and during admission, opioids upon admission, bleeding, and visits to the emergency room. Without a doubt, and in some way, as LeBlanc comments regarding hematologists, “I do my own palliative care” [20], we see that hematologists assume and carry out these tasks, given that the records of the selected variables contain a similar proportion of data. The hematology service should consider delegating some of these tasks. This would result in greater cooperation and reduced team stress.

Odejide [33] analyzes the existence of “a conversation” about goals and care at the end of life between patients and hematologists. It concludes that when these conversations were timely, outpatient, or with hematologists directly involved, patients were less likely to experience heavy healthcare use near death and were more likely to enroll in palliative care. Gebel [34,35], in his studies, also supports this conversation, citing in this case the positive knowledge that patients have about CP without differentiating between ST and HM, and how they expect their oncologists to request these visits. We believe that the existence of this “conversation” is essential for all cancer patients, but in our case, we have not been able to corroborate it because it was not part of the objectives of the study. 

### 4.4. Committee Participation Effects

We saw that there are patients referred to the PCP before they had an AD diagnosed. We think this could be due to the presence of a palliative specialist on the tumor committees as a facilitator of referrals. This was true for one group of tumors (lung cancer, myeloma, and head and neck cancer), but not the others (gynecologic and breast cancer) that did not have palliative care available. The correlation of the recorded symptoms gave us clues to learn more about the nature of pre-AD consultations. We observed that for the first group, with a palliative specialist on its committee, records of physical symptoms (pain, dyspnea) predominated, and in the second group, the record of psycho-emotional symptoms predominated. In the latter case, we believe that the process probably begins at the patient’s own request to their oncologist (Figure 4). In any case, we have observed that the proportion of patients with ST referred to PCP before the definition of AD is higher than to HM (despite the difficulty of defining AD in this group). The ST rate is practically double that of the HM.

### 4.5. Research Implications

Real-world data supports the effectiveness of palliative and psychosocial care services in both groups of pathology. They contradict the belief that these support services negatively interfere with therapeutic decisions and survival (Appendix A). Its exclusive use at the end of life means fewer opportunities for support to resolve outstanding issues and quality of life problems.

We have tried to provide objective data that supports the differences between HM and ST, corroborating that they have different standards. At the same time, these differences can help us reflect on how to improve the relationship between professionals. We advocate for greater collaboration between palliative care and hematology teams to share areas of knowledge and experience.

It would be desirable to integrate PC with the same standardization as physiotherapy or nutrition. From the point of view of quality of care, for the JACIE accreditation of the hematopoietic transplant and cell therapy (HSCT-CT) program in HM, the standards already require the existence of protocols for support services, including CP.

Regarding the clinical factors with significant differences when comparing PCP versus non-PCP in the entire cohort (Table 1), for MH only progression, AD, debut and changes in the therapeutic regimen are maintained. The ECOG has value when performing temporal curves for the entire cohort (Appendix A), but loses it in HM in the subgroup analysis (Appendix A).

Referrals to the ICU (Table 2) maintained significant differences. We see signs that patients referred to PCP are in more situations that are critical or with a worse prognosis, at the end of life. Discussed with our professionals, this situation seems to be a tacit agreement. We must promote a culture of prognostic information and the benefits of early PC and not only consider it at the end of life, allowing holistic support for the special situation of onco-hematology patients.

### 4.6. Generalizability

Temporal stability analysis of the two periods (with and without symptoms data) supports its potential application to similar patient populations. Therefore, these findings should be applicable to cancer centers with established Palliative Care Programs and systematic mortality review processes. And they may not be applicable to health systems without integrated palliative care services or those with different organizational structures (Appendix A).

Minimal differences were observed in the categories of age, PS record, and BMI; we believe they are attributable to the COVD pandemic, not so much because of the associated deaths (in this cohort, it was 3.1%) but because of the changes in the general management of the hospital.

### 4.7. Others Comments 

In the literature, there are models to study the intervals for early referral to PC. One of the studies [2,36] assumes that the consultation is made before 8 weeks of advanced disease (AD), or that it is after 8 weeks of AD. We have not been able to analyze these differences because our study design does not include evolutionary data during the course of PCP or quality of life. 

Real-world data give weight to the good professional work coordinated between support services (both PC and PSOC) with onco-hematology, far from the clichés and beliefs that the participation of these services negatively interferes in therapeutic decisions and is only applicable at the end of life. Specific studies will be necessary that focus attention on the patient and their expectations with measures of quality of life and informed vital decisions (advance directives, preference of place of death and administrative resolutions such as wills), which have not been the scope of study of this presentation. 

It is necessary to find specific situations [20], or screening strategies [34] that justify early consultation with PC.

## 5. Conclusions

### 5.1. Clinical Recommendations 

After the analysis, and due to the prevalence of characteristics or symptoms obtained, we think that the following situations should serve as a warning for early referral to CP:

Patient and tumor-specific related factors: AD, in general, due to either progression or severe debut pathologies: stage IV lung cancer, leukemia, and brain tumors. ECOG pre SACT: III and IV (significant in ST). Recurrent or severe infections (sepsis). Previous ICU admissions. Identification of psycho-emotional symptoms manifested by patients for early referral to PCP.

Symptom-related factors: Present any of the following symptoms: pain, anxiety, sadness, and depression. Need to use opioids. Presence of serious disabling symptoms. 

Other healthcare utilization factors: Need for blood product use (more than 8–15 units) given that polytransfusions indicate worse patient conditions (both physical and psychological–emotional). This suggested number of bags is only indicative and will depend on the type of tumor. In our cohort, for TS, urological cancer used an average of 7 units, followed by gynecological, gastric, and sarcoma. While for HM, patients with leukemia used an average of 25 units, followed by lymphomas and myelomas (Appendix A).

Treatment factors: In patients admitted for intensive treatment and/or hematopoietic stem cell transplant/cellular therapy. This recommendation is supported by bibliographic data [12,26,33,36,37,38,39].

The participation of palliative specialists in the tumor committee facilitates appropriate referrals. Therefore, implement its presence in more tumor committees.

Hematologists historically assume PC tasks, but it is time to improve cooperation to share this burden.

### 5.2. Implementation Considerations

Studies that are more specific are necessary for each type of tumor, especially for the most prevalent types, lung cancer and leukemia, as observed in our cohort, but without leaving aside the psycho-emotional needs demanded by breast and gynecological cancer patients.

It is up to the institutions to evaluate the scope, feasibility, and possible implementation of the suggestions in healthcare practice.

## Figures and Tables

**Figure 1 cancers-17-03467-f001:**
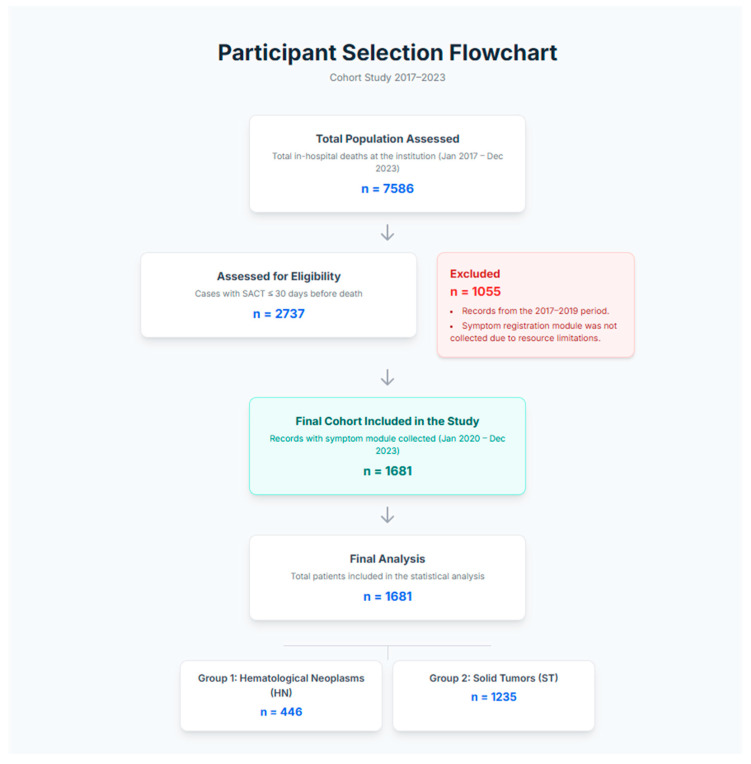
Flow chart shows the participant selection process of study cohort. Note: The cohort selected for this study consists entirely of patients who have information from the new module of symptomatic variables related to palliative care actions. It corresponds to the period 2020–2023. The excluded cases do not have the information and correspond to the oldest period in the database (2017–2019). To find out if there are differences between the periods of the database with or without symptom registration data, see Appendix A (Appendix A), which shows that there are no significant differences in patient characteristics between the two.

**Figure 2 cancers-17-03467-f002:**
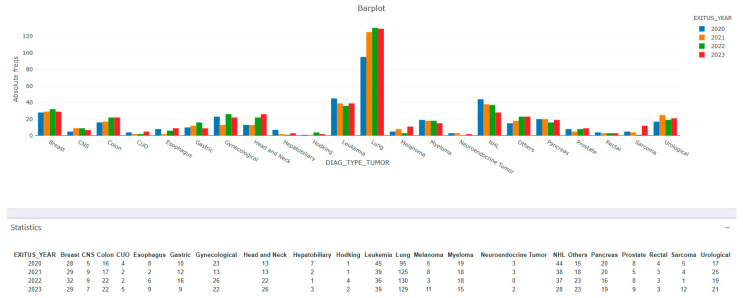
Distribution of tumor pathologies over the years. Note: Distribution of tumor types registered in the MS database. The largest number of cases is lung cancer (28.5%).

**Figure 3 cancers-17-03467-f003:**
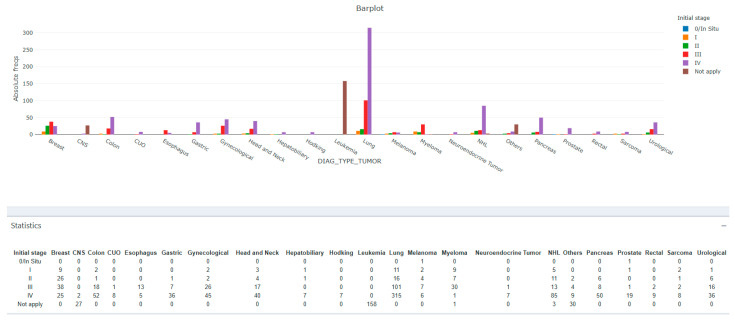
Distribution of tumor pathologies and initial stage. Note: The tumor types grouped under the “not applicable” stage are central nervous system (CNS) tumors, leukemia, and that in “other” include myelodysplastic syndromes (MDS) and lymphoproliferative syndromes (LPS). The majority of cases are stage IV lung cancer, followed by leukemia.

**Figure 4 cancers-17-03467-f004:**
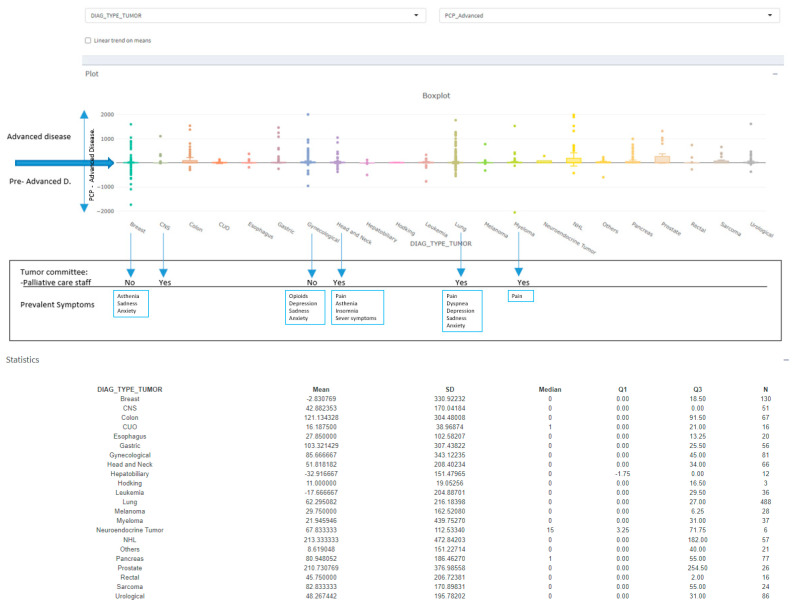
The different periods in which palliative care (PC) operates based on the definition of advanced disease (AD) according to tumor type (excluded cases with follow-up charge, greater than 2000 days). Legend: SD: standard deviation; Q1: first quartile or 25th percentile; Q3: third quartile or 75th percentile; N: number of available values. Note: In the graph, the negative ranges imply that the PC consultations were before the AD definition, we have called them “pre-advanced” visits. On the lowest line, information about the presence or absence of a palliative specialist in the tumor committees (negative for gynecological cancer and breast cancer). In the red boxes, note what type of symptoms are most frequently recorded in “pre-advanced” cases.

**Table 1 cancers-17-03467-t001:** Clinical factors, signs, and symptoms are descriptive of all cohorts and by subgroups of inclusion or not in PCP. Documented new variable registration (log) and the percentage of documented with positive value.

Documented Variable Registration (log) Value Register: Yes	All (1681) N (%)	No PCP (676) N (%)	Yes PCP (1005) N (%)	*p*-Value
CLINICS FACTORS				
SEX (biologic)				0.182
-Male	999 (59.4%)	416 (61.4%)	583 (58.1%)	
-Female	682 (40.6%)	261 (38.6%)	421 (41.9%)	0.182
AGE:				
-current age at death (range:18–98 y)	65.5 (12.3)	67.0 (11.8)	64.6 (12.5)	<0.001
>65 years old	940 (55.9%)	414 (61.2%)	526 (52.4%)	<0.001
ECOG				<0.001
0	70 (7.98%)	46 (13.9%)	24 (4.40%)	
1	390 (44.5%)	164 (49.4%)	226 (41.5%)	
2	321 (36.6%)	96 (28.9%)	225 (41.3%)	
3	86 (9.81%)	24 (7.23%)	62 (11.4%)	
4	10 (1.14%)	2 (0.60%)	8 (1.47%)	
Comorbidity >2	850 (60.3%)	362 (63.3%)	488 (58.2%)	0.065
BODY MASS INDEX (BMI)	25.0 (5.18)	25.9 (5.35)	24.4 (4.97)	<0.001
BMI by categories:				0.031
-Underweight	41 (6.29%)	13 (4.89%)	28 (7.25%)	
-Normal weight	361 (55.4%)	133 (50.0%)	228 (59.1%)	
-Overweight	169 (25.9%)	77 (28.9%)	92 (23.8%)	
-Obesity	74 (11.3%)	39 (14.7%)	35 (9.07%)	
-Morbid obesity	7 (1.07%)	4 (1.50%)	3 (0.78%)	
PROGESSION DISEASE	993 (59.6%)	335 (50.0%)	658 (66.1%)	<0.001
ADVANCED DISEASE (AD)	1308 (77.8%)	445 (65.7%)	863 (86.0%)	<0.001
SCHEMA THERAPEUTIC:				
-Debut	351 (20.9%)	180 (26.6%)	171 (17.0%)	<0.001
-Change	369 (22.0%)	132 (19.5%)	237 (23.6%)	0.053
-Continuation	823 (49.0%)	307 (45.3%)	516 (51.4%)	0.017
TRIAL	135 (8.07%)	60 (8.90%)	75 (7.51%)	0.349
LEVEL THERAPEUTIC INTERVENTION (LTI)				
-LTI informed at admission	1372 (82.5%)	499 (74.9%)	873 (87.5%)	<0.001
-LTI coherence intervention	1017 (60.5%)	312 (46.1%)	705 (70.2%)	<0.001
-ICU admission	196 (11.7%)	155 (22.9%)	41 (4.08%)	<0.001
-Intubation	46 (2.74%)	37 (5.47%)	9 (0.90%)	<0.001
LAST MONTH EMERGENCIES	1390 (82.7%)	563 (83.2%)	827 (82.4%)	0.561
SINGS/SYMPTOMS				
PAIN log	1661 (98.8%)	660 (97.5%)	1001 (99.7%)	<0.001
Pain: Yes	1404 (83.6%)	532 (78.8%)	872 (86.9%)	<0.001
PRIOR ADMISION_ANALGESIA log	1427 (84.9%)	520 (76.8%)	907 (90.3%)	<0.001
Prior analgesia: Yes	1291 (77.9%)	455 (68.5%)	836 (84.1%)	<0.001
ANALGESIA_ON ADMISION log	1278 (76.0%)	493 (72.8%)	785 (78.2%)	0.014
Analgesia on admission: Yes	1181 (72.1%)	454 (69.3%)	727 (74.0%)	0.084
PRIOR ADMISION _OPIOIDS log	1039 (61.8%)	308 (45.5%)	731 (72.8%)	<0.001
Prior opioids: Yes	837 (51.2%)	214 (32.8%)	623 (63.4%)	<0.001
OPIOIDS_ON ADMISION log	1304 (77.6%)	485 (71.6%)	819 (81.6%)	<0.001
Opioids on admission: Yes	1241 (76.1%)	451 (69.2%)	790 (80.8%)	<0.001
DYSPNEA log	1581 (94.1%)	618 (91.3%)	963 (95.9%)	<0.001
Dyspnea Yes	1073 (64.2%)	430 (64.1%)	643 (64.2%)	0.001
ASTHENIA log	1491 (88.7%)	546 (80.6%)	945 (94.1%)	<0.001
Asthenia Yes	1437 (86.0%)	520 (77.4%)	917 (91.8%)	<0.001
INSOMNIA log	936 (55.7%)	240 (35.5%)	696 (69.3%)	<0.001
Insomnia Yes	764 (45.8%)	203 (30.3%)	561 (56.2%)	<0.001
DEPRESSION log	448 (26.7%)	148 (21.9%)	300 (29.9%)	<0.001
Depression Yes	321 (19.3%)	106 (15.9%)	215 (21.7%)	0.002
ENDICOTT TEST log	292 (17.4%)	16 (2.36%)	276 (27.5%)	<0.001
ENDICOTT Positive	24 (1.45%)	0 (0.00%)	24 (2.44%)	<0.001
SADNESS log	645 (38.4%)	110 (16.2%)	535 (53.3%)	<0.001
Sadness Yes	547 (33.0%)	98 (14.7%)	449 (45.3%)	<0.001
ANXIETY log	947 (56.3%)	260 (38.4%)	687 (68.4%)	<0.001
Anxiety Yes	766 (46.0%)	217 (32.4%)	549 (55.2%)	<0.001
SEVERE SYMPTOMS log	1225 (72.9%)	474 (70.0%)	751 (74.8%)	0.035
Severe Symptoms Yes	541 (35.5%)	200 (33.5%)	341 (36.9%)	0.377
SEVERE SYMPTOM TYPES:				0.028
-Brain involvement	215 (40.1%)	71 (36.2%)	144 (42.4%)	
-Carcinomatosis and effusion (Pleural pericardial or peritoneal)	111 (20.73%)	37 (18.85%)	74 (21.78%)	
-Spine Compression	49 (9.14%)	12 (6.12%)	37 (10.9%)	
-Severe acute respiratory failure	30 (5.60%)	15 (7.65%)	15 (4.41%)	
-Carcinomatous lymphangitis	14 (2.61%)	6 (3.06%)	8 (2.35%)	
-Septic Shock	9 (1.68%)	5 (2.55%)	4 (1.18%)	
-Acute hemorrhage	16 (2.99%)	10 (5.10%)	6 (1.76%)	
-Vertebral crush (M1)	7 (1.31%)	1 (0.51%)	6 (1.76%)	
-Vena Cava Compression Synd.	3 (0.56%)	2 (1.02%)	1 (0.29%)	
-Acute lung edema	1 (0.19%)	1 (0.51%)	0 (0.00%)	
-Others	81 (15.1%)	36 (18.4%)	45 (13.2%)	
TRANSFUSIONS log	1461 (86.9%)	576 (85.1%)	885 (88.1%)	0.079
Transfusions Yes	998 (60.0%)	412 (61.4%)	586 (59.0%)	0.014
TRANSFUSED BAGS (average—SD)	9.35 (16.0)	12.7 (19.6)	6.91 (12.1)	<0.001
INFECTION log	1492 (88.8%)	595 (87.9%)	897 (89.3%)	0.397
Infection Yes	971 (58.7%)	441 (66.4%)	529 (53.3%)	<0.001
HEMORRHAGE log	1169 (69.5%)	454 (67.1%)	715 (71.2%)	0.078
Hemorrhage Yes	393 (24.0%)	176 (26.7%)	215 (21.9%)	0.009
LABORATORY:				
-Hemoglobin (gr/dL)	10.6 (2.01)	10.4 (2.13)	10.8 (1.92)	0.004
-Platelets (×10^9^/L)	253 (310)	222 (325)	272 (298)	0.005
-Albumin (gr/L)	33.8 (7.22)	33.2 (7.38)	34.1 (7.11)	0.148
-LDH (ukat/Ll)	512 (712)	534 (720)	496 (706)	0.571
PSICHO-SOCIAL PROGRAM (PSOP)	1128 (67.3%)	124 (18.5%)	1004 (100%)	<0.001

Legend: Categorical variables: Frequencies (Percentage, %). Normally distributed numerical variables: Average (standard deviation). Non-normally distributed numerical variables: Median [25th; 75th percentiles]. If the *p*-value < 0.05 means that, the differences or proportions between the groups are statistically significant with a significance level of 5%. SD: standard deviation. Note: PCP: Palliative Care Program; log: any record regarding the symptom documented in electronic register; Yes: when the record indicated the positivity of the symptom. For details in transfused bags by tumor types, see (Appendix A).

**Table 2 cancers-17-03467-t002:** Descriptive and comparative by pathology group (ST vs. HM) and referral or not to palliative care (“PCP No” vs. “PCP Yes”).

Summary Descriptive	Solid Tumor by Groups of ‘PCP’			Hematological Malignances by Groups of ‘PCP’		
	PCP No	PCP Yes	*p*-Value	PCP No	PCP Yes	*p*-Value
	N = 393 (31.8%)	N = 842 (68.2%)		N = 284 (63.7%)	N = 162 (36.3%)	
PAIN log	387 (98.5%)	841 (99.9%)	0.005	273 (96.1%)	160 (98.8%)	0.147
Prior Analgesia log	306 (77.9%)	769 (91.3%)	<0.001	214 (75.4%)	138 (85.2%)	0.020
Analgesia on admission log	266 (67.7%)	660 (78.4%)	<0.001	227 (79.9%)	125 (77.2%)	0.569
Prior Opioids log	187 (47.6%)	624 (74.1%)	<0.001	121 (42.6%)	107 (66.0%)	<0.001
Opioids on admission log	290 (73.8%)	702 (83.4%)	<0.001	195 (68.7%)	117 (72.2%)	0.496
Dyspnea log	360 (91.6%)	807 (95.8%)	0.004	258 (90.8%)	156 (96.3%)	0.051
Asthenia log	322 (81.9%)	795 (94.4%)	<0.001	224 (78.9%)	150 (92.6%)	<0.001
Insomnia log	170 (43.3%)	603 (71.6%)	<0.001	70 (24.6%)	93 (57.4%)	<0.001
Depression log	91 (23.2%)	252 (29.9%)	0.016	57 (20.1%)	48 (29.6%)	0.030
ENDICOTT log	12 (3.05%)	232 (27.6%)	<0.001	4 (1.41%)	44 (27.2%)	<0.001
Sadness log	71 (18.1%)	462 (54.9%)	<0.001	39 (13.7%)	73 (45.1%)	<0.001
Anxiety log	155 (39.4%)	589 (70.0%)	<0.001	105 (37.0%)	98 (60.5%)	<0.001
Severs Symptom log	270 (68.7%)	643 (76.4%)	0.005	204 (71.8%)	108 (66.7%)	0.300
Transfusion log	327 (84.1%)	741 (88.9%)	0.006	249 (88.3%)	145 (89.5%)	0.843
Transfused bags (*)	6.39 (11.1)	9.29 (16.5)	0.040	14.7 (20.4)	6.95 (10.6)	0.002
Infections log	340 (86.5%)	754 (89.5%)	0.143	255 (89.8%)	143 (88.3%)	0.735
Hemorrhages log	242 (61.6%)	604 (71.7%)	<0.001	212 (74.6%)	111 (68.5%)	0.200
Advanced disease	316 (80.4%)	769 (91.3%)	<0.001	129 (45.4%)	94 (58.0%)	<0.001
Intubation	9 (2.29%)	6 (0.71%)	0.025	28 (9.86%)	3 (1.85%)	0.003
ICU: admission ICU: death	62 (15.8%) 52 (13.3%)	37 (3.44%) 20 (2.38%)	<0.001	122 (43.0%) 103 (36.5%)	12 (7.41%) 9 (5.59%)	<0.001
Resuscitation	1 (0.25%)	1 (0.12%)	1.000	---	---	---
Last month emergencies	363 (92.4%)	703 (83.5%)	<0.001	200 (70.4%)	124 (76.5%)	0.520
If visits: How many visits?			0.911			0.780
1	221 (61.7%)	424 (61.0%)		134 (68.0%)	84 (68.9%)	
2	105 (29.3%)	212 (30.5%)		49 (24.9%)	27 (22.1%)	
3	30 (8.38%)	49 (7.05%)		13 (6.60%)	11 (9.02%)	
4	1 (0.28%)	5 (0.72%)		1 (0.51%)	0 (0.00%)	
5	1 (0.28%)	3 (0.43%)		---	---	
6	0 (0.00%)	1 (0.14%)		---	---	
7	0 (0.00%)	1 (0.14%)		---	---	
N/D	5 (1.37%)	8 (1.13%)		3 (1.50%)	2 (1.61%)	
PCP < 3 d	---	243 (28.9%)	---	---	49 (30.4%)	---
PSOP	67 (17.0%)	842 (100%)	<0.001	57 (20.4%)	162 (100%)	<0.001

Note: Log: reference to cases in which there is some type of record of the variable (either the record of positive or negative data) in the patient’s electronic history. (*) For more details on the number of bags transfused according to tumor type, see Appendix A.

## Data Availability

The majority of original contributions presented in this study are included in the article and Appendix A. Further inquiries for raw data can be directed to the corresponding author.

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
