# Peer review of "Role of Palliative Care in Onco-Hematology Retrospective Observational Cohort Study in Deceased In-Hospital Patients with SACT at the End of Life: Experience with Real-World Data from a Cancer Monographic Institution"

_cancers, 2025, doi:10.3390/cancers17213467_

Round 1
Reviewer 1 Report
Comments and Suggestions for Authors
Major revisions required
-
Title and Abstract
-
The title suggests an interventional approach (“How to improve participation...”), while the study is observational. Please revise to reflect the actual design.
-
In the abstract, methods should be presented more clearly. Avoid spoken language (e.g., “we asked ourselves...”).
-
Use “referral to palliative care” rather than “participation,” which is more precise.
-
Numbers are inconsistent: the abstract mentions 1,681 patients (61% of database) while Methods states 2,737 cases (36% of deaths). This discrepancy must be resolved and explained consistently.
-
-
Objectives and Consistency
-
The stated aim (“to identify variables that help improve early indication...”) does not fully align with the descriptive comparisons (PCP vs. non-PCP, ST vs. HN). Please reformulate the aim to match the study design and outcomes.
-
Recommendations listed in the abstract and conclusion (e.g., lung cancer stage IV, leukemia, transfusion thresholds, disabling symptoms) appear too specific relative to the presented data and should be revised for consistency.
-
-
Methods
-
The inclusion period is confusing. The study covers 2017–2023, but only the 2020–2023 period is fully analyzed. This is only explained later in the flowchart legend, due to missing palliative care variables in earlier years. This critical point must be clearly described at the beginning of the Methods section to avoid reader confusion.
-
The decision to compare the two periods (2017–2019 vs. 2020–2023) should also be explained upfront, not introduced retrospectively.
-
Flowchart information should be integrated into the Methods text before describing participants.
-
The list of new variables (pain, opioid use, dyspnea, insomnia, depression screening, “Endicott test,” sadness, etc.) needs clearer justification. Why was “sadness” included separately from validated measures such as depression or anxiety? Please clarify operational definitions and their clinical rationale.
-
-
Results
-
Ensure consistency between main text and Supplementary Tables (percentages and p-values).
-
The key comparison (ST vs. HN) remains descriptive; multivariate analyses would strengthen conclusions about independent predictors of PCP referral.
-
-
Discussion
-
The discussion should be expanded to engage more deeply with international guidelines (ASCO, ESMO, NCCN).
-
Limitations (retrospective design, incomplete early dataset, exclusion of out-of-hospital deaths, absence of psychosocial variables) need to be explicitly acknowledged.
-
Recommendations should be rephrased to align directly with the actual findings and study aim.
-
Minor revisions
-
Harmonize abbreviations (ST, HN, PCP, SACT).
-
Correct spacing/typos (e.g., “compareGroups” → “compare Groups”).
-
Remove spoken or informal phrasing (“Finally, we asked ourselves…”).
-
Adjust punctuation in Introduction (e.g., WHO definition sentence).
The English is generally understandable but requires substantial editing for grammar, style, and clarity.
-
Several sentences are too long or colloquial.
-
Abstract and introduction contain spoken-language formulations.
-
Some terms are inconsistent (e.g., “participation” vs. “referral”).
→ A thorough professional language revision is recommended.
Author Response
Major revisions required
1. Title and Abstract
- The title suggests an interventional approach (“How to improve participation...”), while the study is observational. Please revise to reflect the actual design.
- In the abstract, methods should be presented more clearly. Avoid spoken language (e.g., “we asked ourselves...”).
We have simplified the methods in the abstract.
We have removed the sentences with spoken English.
- Use “referral to palliative care” rather than “participation,” which is more precise.
I used the term "participation" to avoid excessive repetition of words, but I agree that "referral" is more accurate in this content and have modified it in the text.
- Numbers are inconsistent: the abstract mentions 1,681 patients (61% of database) while Methods states 2,737 cases (36% of deaths). This discrepancy must be resolved and explained consistently.
We wanted to describe the entire database to show the difficulty of the retrospective study. As you allege in your comment, I see that it is not well understood, it does not contribute anything to the results and generates confusion. We have deleted that refers to the period not included in the study, left only the minimum to reference the origin of the period included in the study. The rest of the information will be available in supplementary material.
2. Objectives and Consistency
- The stated aim (“to identify variables that help improve early indication...”) does not fully align with the descriptive comparisons (PCP vs. non-PCP, ST vs. HN). Please reformulate the aim to match the study design and outcomes.
We have reformulated the main objective of the study:
“The main objective is to know what role CP plays in supporting each group of pathologies. The indirect indicator of CP activity is the measure of the percentage of symptoms identified in electronic records (study with real-world data)”.
- Recommendations listed in the abstract and conclusion (e.g., lung cancer stage IV, leukemia, transfusion thresholds, disabling symptoms) appear too specific relative to the presented data and should be revised for consistency.
It is true that some of the recommendations do not seem to have an explanation for their support. We have added the following phrase to draw attention to the origin of its obtaining.
“According to the prevalence obtained, and bibliographic support, we intend to make some recommendations for the referral of these patients to CP”
3. Methods
- The inclusion period is confusing. The study covers 2017–2023, but only the 2020–2023 period is fully analyzed. This is only explained later in the flowchart legend, due to missing palliative care variables in earlier years. This critical point must be clearly described at the beginning of the Methods section to avoid reader confusion.
As suggested to us, we have detailed the cohort selection in the text, which is located at the bottom of Fig. 1.
- The decision to compare the two periods (2017–2019 vs. 2020–2023) should also be explained upfront, not introduced retrospectively.
The comparison of characteristics between both periods can be consulted in complementary material (S1), because we consider that since there are no significant differences between them, the conclusions could be extended to all registered patients.
- Flowchart information should be integrated into the Methods text before describing participants.
As you suggest the flowchart is found after the description of the patient material and methods.
- The list of new variables (pain, opioid use, dyspnea, insomnia, depression screening, “Endicott test,” sadness, etc.) needs clearer justification. Why was “sadness” included separately from validated measures such as depression or anxiety? Please clarify operational definitions and their clinical rationale.
The choice of sadness was at the request of our palliative care specialists as an element not always linked to depression or anxiety and because they recognize that they raise this issue with our patients
4. Results
- Ensure consistency between main text and Supplementary Tables (percentages and p-values).
I have revised the results so that they are consistent in the text and tables
- The key comparison (ST vs. HM) remains descriptive; multivariate analyses would strengthen conclusions about independent predictors of PCP referral.
Once again we agree that a multivariate analysis would strengthen the results.
But we have not included a multivariate analysis because we designed the study as a descriptive observational analysis, focused on descriptive comparisons for methodological reasons: description of the differences between referrals to PCP and between pathologies (TS vs HM) in the real world and the retrospective nature.
We have included this annotation in material and methods
5. Discussion
- The discussion should be expanded to engage more deeply with international guidelines (ASCO, ESMO, NCCN).
We have improved the discussion taking into account international guidelines.
- Limitations (retrospective design, incomplete early dataset, exclusion of out-of-hospital deaths, absence of psychosocial variables) need to be explicitly acknowledged.
We have included them in section 2.5 Material and method biases
All consecutive cases from the 2020-2023 period met the criteria and were included, thus minimizing selection bias.
The retrospective nature of data collection from electronic records can introduce biases in the information; therefore, the trained personnel who performed the data entry manually work under medical supervision.
The fact that all cases correspond exclusively to hospital deaths could also represent a bias. In this sense, we accept that our indicators may have values ​​higher than those published by other authors. Hospitalization indicates a condition of greater severity or risk for patients.
The definition of AD in hematology is more complex than in ST. This is especially true in the case of leukemia [28]. This complexity also partially limited our ability to obtain AD-PCP intervals for all HM.
The inclusion of all ages, all causes of death, patients with advanced and non-advanced disease may represent a bias in comparison with other published data.
Clinical data comes from electronic medical records of real patients. This is not data from epidemiological records or administrative billing, nor synthetic data.
Resource limitations have influenced the lack of continuous monitoring of signs and symptoms and, therefore, the evaluation of the appropriate moment for referral to palliative care. It has also limited the ability to establish the correlation with psychosocial and quality of life variables, which is why we have left them out of our scope of study.
The new variable registration only considers whether the variable is registered in the electronic medical record of each patient. Therefore, we assume that we only have an indirect measure of PCP activity.
Based on the results obtained, our recommendations have to do with the most prevalent groups in the cohort and the differences in the recording of symptoms (and other clinical factors) between patients referred to palliative care and those not referred.
- Recommendations should be rephrased to align directly with the actual findings and study aim.
We have included them in section 5.1 of conclusions.
Minor revisions
- Harmonize abbreviations (ST, HN, PCP, SACT).
As suggested, we have reviewed and Harmonized the abbreviations
- Correct spacing/typos (e.g., “compareGroups” → “compare Groups”).
“compareGroups”: Descriptive Analysis by Groups . It’s de name of de statistical package source.
- Remove spoken or informal phrasing (“Finally, we asked ourselves…”).
As suggested, we have revised and deleted the informal English phrases
- Adjust punctuation in Introduction (e.g., WHO definition sentence).
As suggested, we have improved the in-text citations.
Comments on the Quality of English Language
The English is generally understandable but requires substantial editing for grammar, style, and clarity.
- Several sentences are too long or colloquial.
- Abstract and introduction contain spoken-language formulations.
- Some terms are inconsistent (e.g., “participation” vs. “referral”).
→ A thorough professional language revision is recommended.
We appreciate your clarification about the level of written English.
We have requested a new revision of the language.
Reviewer 2 Report
Comments and Suggestions for Authors
While the data presented in this manuscript show some interesting insight (differences regarding involvement of palliative care between solid tumour and haematologic malignancies; degree of systematic registration of symptoms; differences in management of severe symptoms with/without palliative care regarding ICU admissions, transfusions, visits to emergency department), the article itself requires a substantial makeover before publication. The overall readability is poor and there are severe linguistic flaws (spelling, grammar, use of tenses, syntax). This is also the case for the inscriptions in the figures.
With regard to figure 3, I don't really see any added value to those results. Giving pooled survival functions for solid vs haematologic malignancies is not a valid comparison and doesn't add anything to your findings.
The clinical recommendations with factors for early involvement of palliative care that you list in the conclusions are interesting! However, it must be stated more clearly how you extract those factors from the data you extracted.
I would recommend to also focus in the discussion on the points that I pointed out as valuable insight in the very first paragraph of my review.
Overall, your data has good potential but needs a better presentation and focus to warrant publication.
Author Response
While the data presented in this manuscript show some interesting insight (differences regarding involvement of palliative care between solid tumour and haematologic malignancies; degree of systematic registration of symptoms; differences in management of severe symptoms with/without palliative care regarding ICU admissions, transfusions, visits to emergency department), the article itself requires a substantial makeover before publication. The overall readability is poor and there are severe linguistic flaws (spelling, grammar, use of tenses, syntax). This is also the case for the inscriptions in the figures.
Thank you very much for your thoughtful comments. We have done a complete language review by an English speaking colleague.
With regard to figure 3, I don't really see any added value to those results. Giving pooled survival functions for solid vs haematologic malignancies is not a valid comparison and doesn't add anything to your findings.
We agree with him in his comment.
It is true that the survival curves between ST and HM are not at all comparable. We wanted to emphasize the results of HM since obtaining worse indicators is not indicative of malpractice, but rather of a different disease that deserves better considerations. We have removed the figure from the text.
The clinical recommendations with factors for early involvement of palliative care that you list in the conclusions are interesting! However, it must be stated more clearly how you extract those factors from the data you extracted.
We have improved the explanations of the recommendations, they are included in section 5.1 of conclusions.
I would recommend to also focus in the discussion on the points that I pointed out as valuable insight in the very first paragraph of my review.
We have expanded the discussion by trying to review all the points you refer to. There is also supplementary material that supports the data collected.
Overall, your data has good potential but needs a better presentation and focus to warrant publication.
We thank you for all your suggestions and we hope we have been able to correct them. We remain at your disposal if further improvements are necessary.